# Construct prognostic models of multiple myeloma with pathway information incorporated

Shuo Wang[1,2,3], ShanJin Wang[1], Wei Pan[1], YuYang Yi[1], Junyan Lu[3]*

**1** Department of Spinal Surgery, Shanghai East Hospital, School of Medicine, Tongji University, Shanghai, China, **2** Institute of Medical Biometry and Statistics, Faculty of Medicine and Medical Center—University of Freiburg, Freiburg, Germany, **3** Institute of Computation Biomedicine and Center for Infectiology, Medical Faculty Heidelberg, Heidelberg University, Heidelberg, , Germany

☯ These authors contributed equally to this work.
* junyan.lu@uni-heidelberg.de

**Data Availability Statement:** All the gene expression data and clinical information used in this study are publicly available on Gene Expression Omnibus (GEO) with the accession IDs: GSE136324, GSE2658 and GSE9782. The codes

## Abstract

Multiple myeloma (MM) is a hematological disease exhibiting aberrant clonal expansion of cancerous plasma cells in the bone marrow. The effects of treatments for MM vary between patients, highlighting the importance of developing prognostic models for informed therapeutic decision-making. Most previous models were constructed at the gene level, ignoring the fact that the dysfunction of the pathway is closely associated with disease development and progression. The present study considered two strategies that construct predictive models by taking pathway information into consideration: pathway score method and group lasso using pathway information. The former simply converted gene expression to sample-wise pathway scores for model fitting. We considered three methods for pathway score calculation (ssGSEA, GSVA, and z-scores) and 14 data sources providing pathway information. We implemented these methods in microarray data for MM (GSE136324) and obtained a candidate model with the best prediction performance in interval validation. The candidate model is further compared with the gene-based model and previously published models in two external data. We also investigated the effects of missing values on prediction. The results showed that group lasso incorporating Vax pathway information (Vax(grp)) was more competitive in prediction than the gene model in both internal and external validation. Immune information, including VAX pathways, seemed to be more predictive for MM. Vax (grp) also outperformed the previously published models. Moreover, the new model was more resistant to missing values, and the presence of missing values (<5%) would not evidently deteriorate its prediction accuracy using our missing data imputation method. In a nutshell, pathway-based models (using group lasso) were competitive alternatives to gene-based models for MM. These models were documented in an R package (https://github.com/ShuoStat/MMMs), where a missing data imputation method was also integrated to facilitate future validation.

for reproducing all the analysis results in this study are provided on GitHub (https://github.com/ ShuoStat/MMModels). Our proposed models are implemented in the package (MMMs), which can be accessed via GitHub (https://github.com/ ShuoStat/MMMs).

**Funding:** This work was financially supported by the German Federal Ministry of Education and Research (SMART-CARE under grant agreement 161L0213, to JL), the Training Program for Academic and Technical Leaders of Major Disciplines in Jiangxi Province-Leading Talents Project (20213BCJ22011 to SJW), Key Projects of Natural Science Foundation of Jiangxi Province (20212ACB206032 to SJW) and Discipline Leader Training Plan of Pudong New Area Health Commission (PWRd2020-11 to SJW). The funders had no role in study design, data collection and analysis, decision to publish, or preparation of the manuscript.

## Author summary

Traditionally, prognostic models were mainly constructed at the gene level, ignoring the role of pathway functions in disease development and progression. Enlightened by this, we advocated guiding the model building with well-established prior knowledge (pathway information). We investigated several approaches that could incorporate pathway information in model building. The results showed that pathway-based models exhibit superior predictive capabilities compared to their gene-based counterparts. Furthermore, these pathway-based models were more robust to missing values. The proposed models also outperformed previously published gene-based models. Beyond their prediction performance, the pathway-based models can directly reveal the association between the pathway functions and survival outcomes, demonstrating their advantage in the model interpretability. This enhanced interpretability not only deepens our understanding of disease mechanisms but also facilitates informed decision-making in clinical and research settings. We urge for more attention to be given to developing modeling methods that incorporate prior knowledge.

## 1 Introduction

Multiple myeloma (MM) is a hematological disease exhibiting aberrant clonal expansion of cancerous plasma cells in bone marrow [1–3]. Multiple myeloma (MM) usually starts from monoclonal gammopathy of undetermined significance (MGUS), progresses to symptomatic multiple myeloma (SMM), and ultimately develops into MM. Approximately 1% of MGUS and 10% of SMM will progress to MM each year [4,5]. Many treatments have been approved for MM, but the prognosis varies between patients [6,7]. Predictive models can help distinguish the patients of high risk for therapeutic decision-making [8]. Several models have been proposed either based on clinical information or based on gene expression data [9–11]. However, few of them have been applied in practice for several reasons, including insufficient predictive accuracy, lack of generalization to the whole population, and poor model reporting [10,12,13].

Most previous models were constructed at the gene level. Indeed, the development of disease involves the dysfunction of biological pathways, where clusters of genes collaborate to perform specific functions [14]. Hence, incorporating pathway information in model building can provide more accurate representations of disease progression [15]. In addition, gene-based models (particularly sparse models) encounter challenges when applied to other datasets due to genetic heterogeneity and measurement errors [8]. However, pathway-based models are considered to be more robust to uncontrolled variation as they incorporate well-established prior knowledge which can significantly reduce the variance caused by noise variables [16,17,11]. Furthermore, pathway-based models are usually more interpretable due to their inherent biological relevance [8,18]. The selected pathway models can directly reflect the relevance between biological functions (instead of individual genes) and survival outcomes. Moreover, these selected pathways can be the clues guiding the research of biological mechanisms and precise medicine.

In this light, the present study explored two possible strategies that incorporated pathway information in model building: pathway score methods (including ssGSEA, GSVA, and z-score) and group lasso. We implemented these methods in several MM data, investigating their properties and comparing their prediction performance with the gene-based model and previously published models. Our benchmark results demonstrated clear advantage of

pathway-based models in clinical outcome prediction. Our computational workflow can also serve as a comprehensive guidebook for future studies where prior knowledge needs to be incorporated into predictive models.

## 2 Methods

### 2.1 Datasets

We used three microarray datasets derived from Gene Expression Omnibus (GEO, https://www.ncbi.nlm.nih.gov/geo/), with the data of the largest effective sample size being the training data (GSE136324), and the rest two being the test data (GSE9782 and GSE2658). All three data were derived from prospective studies with overall survival by months as endpoints. Probes were removed if i) they did not target any gene symbol or targeted multiple symbols; ii) the targeted genes were not available in all three datasets because the external validation required the training data and test data to have the same data structure. Finally, we included 11,485 genes.

GSE136324 enrolled patients received the Total Therapy (TT) 3–5 during 2004–2014 [4,19–21]. It contains 867 samples from 436 patients, out of which 298 were repeatedly measured (Table 1). The initial measurements were used for the patients who were repeatedly measured. This process yielded 436 samples, including 266 males and 170 females. The follow-up time ranged from 0 to 174 months (median 92 months), and events were observed in 191 samples (43.8%). The gene expression profile was normalized with MAS5.0 and transformed into log2 scale. No missing value was observed.

GSE9782 was generated from a multiple-centered clinical trial of bortezomib covering 12 countries [22]. It enrolled 669 relapsed patients who received one to three prior therapies, of which 264 microarray profiles were accessible (159 males and 105 females). The follow-up time ranged from 2 to 37 months (median 15 months). GSE2658 contained 351 newly diagnosed MM cases [23,24,3]. The follow-up ranged from 0 to 69 months (median 23 months), during which 100 events (17.9%) were observed.

### 2.2 Workflow for model building and validation

We considered two strategies for model building that incorporate pathway information, namely pathways score method and group lasso. For the former, we employed three methods (ssGSEA, GSVA, and z-score) to calculate pathway scores. These methods would be implemented in 14 pathway databases and compared to the gene-based model. The prediction performance was evaluated both in internal validation (within the same data) and in external validation (GSE9782 and GSE2658). The purpose of the internal validation was to assess the prediction performance of the models in the data of the same population, while the external

Table 1. Baseline information of the three datasets.

|  | GSE136324 | GSE9782 | GSE2658 |
|---|---|---|---|
| Sample size | 436 | 264 | 559 |
| Gender (M/ F) | 266 / 170 | 159 / 105 | / |
| Age | 58.5(9.1) | 60.2(10.2) | / |
| Number of Events (%) | 191 (43.8%) | 156 (59.1%) | 100 (17.9%) |
| Ranges of follow-up time | 0–174 | 2–37 | 0–69 |
| Patients | Newly diagnosed | Relapsed | Newly diagnosed |
| Where to use | Training models; Internal validation | External validation | External validation |

validation could assess the reliability of models on the new data probably representing different populations.

The workflow of this paper is shown in Fig 1.

## 2.3 Pathway score methods

The pathway score method simply fits models using pathway scores, which are converted from gene expressions based on pre-defined pathway information (e.g., hallmark pathways). We considered employing the least absolute shrinkage and selection operator (Lasso) for model building. More advanced machine learning methods, such as random forest, gradient boosting, SVM, and so on, or deep learning methods, can also be used for incorporating pathway

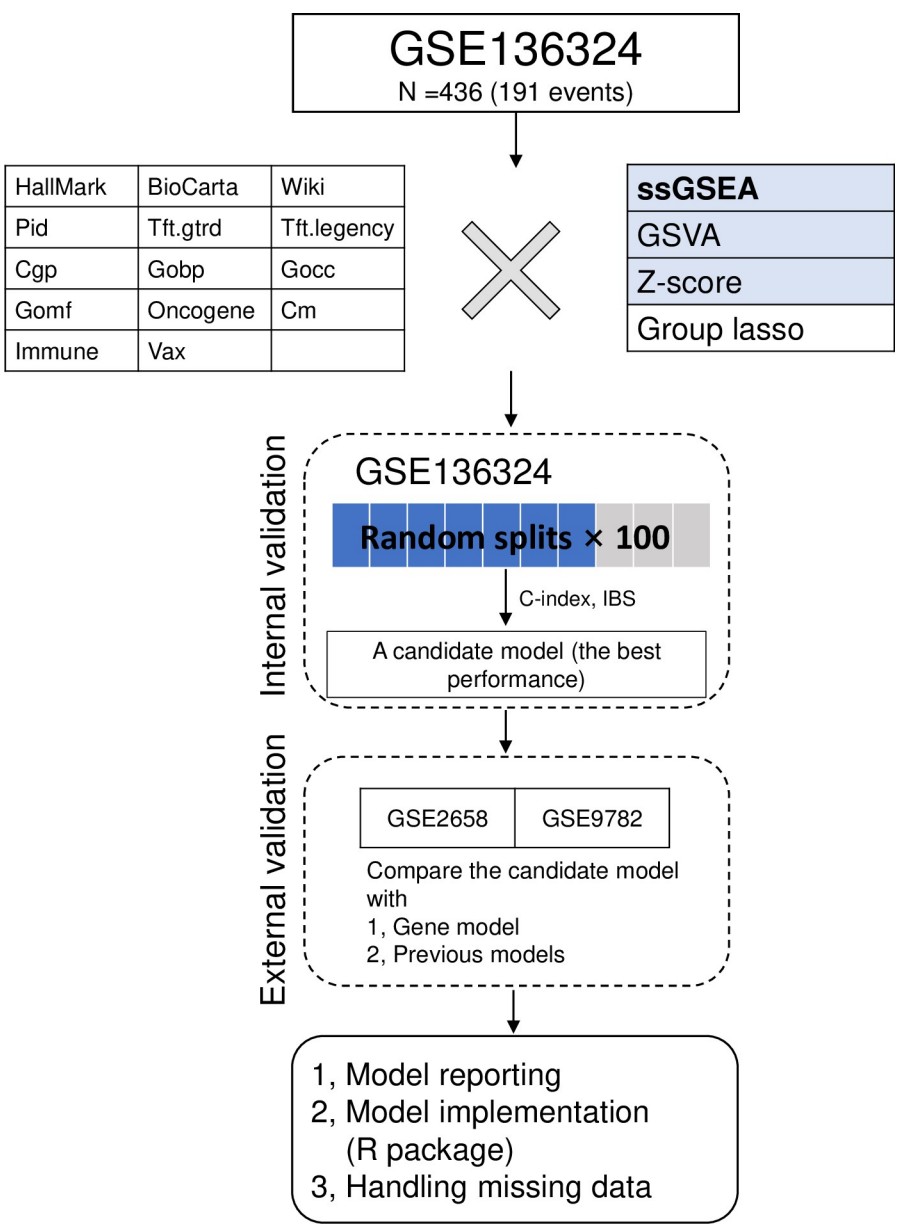

**Fig 1. The workflow of the model fitting and validation procedure.**

information into prediction models and may have better performance than simple multiple linear regression. However, we prefer multiple linear regression with Lasso penalty because of its simplicity and interpretability. More sophisticated machine learning or deep learning methods typically require large sample sizes and often face challenges in model interpretability, and therefore not ideal for clinical applications. Nevertheless, our current study in a simple context demonstrates the principles and benefits of incorporating pathway information in outcome prediction and opens up the possibility of building and testing more advanced machine learning/AI methods.

**2.3.1 Calculate pathway scores.** Several methods have been proposed to calculate pathway scores. [25] proposed GSVA, a nonparametric and unsupervised method to measure the sample-wise pathway activities. [26] proposed Pathifier that measures the distance of a case projected on the principal curve to a reference point (typically the centroid of a reference set). It was further used by [17,11,27]. [16] employed the principal components analysis (PCA) to capture the main information of pathways. [28] defined the pathway score as the prognostic risk of the pathway by fitting the penalized Cox model using all predictors within the pathway.

In this study, we consider three methods for the calculation of pathway scores, including the single-sample GSEA (ssGSEA), GSVA, and z-scores [29]. Out of them, ssGSEA and GSVA have been widely used in Bioinformatics [29,25]. Z-score was selected for its simplicity. Additionally, each of these methods has distinct features. For instance, ssGSEA relies on rank information, and its calculation does not depend on background samples. In contrast, GSVA and Z-score are representative methods that depend on the distribution of all samples in the cohort, and therefore require a reference cohort if scores need to be calculated for new samples. Other scoring methods, such as PCA and Pathifier, were not used in the current study for various reasons. PCA is very sensitive to extreme values and choosing which PC for pathway score is not a trivial task. Pathifier is computationally intensive and requires parameter tuning and therefore not suitable for large-scale benchmarking and clinical application. In addition, Pathifier has also been shown to be sensitive to outliers.

## ssGSEA

ssGSEA is a pathway activity measurement based on gene ranks of individual samples [30]. It measures the rank distance between a predefined gene set and the rest of the genes. ssGSEA has two favorable features: i) the calculation of ssGSEA entirely relies on gene ranks instead of gene expression values, which is helpful to mitigate batch effects and measurement errors; ii) the calculation of ssGSEA does not depend on background samples as it merely uses the information of gene ranks. These two features are desirable in practical application because the established model can be easily used for an individual patient without complex data processing (e.g., normalization) that requires background samples.

## GSVA

Unlike ssGSEA which directly orders genes by expression levels, GSVA calculates the cumulative density function to all other cases after a kernel transformation [25]. The total density is transformed into ranks, which were subsequently used to calculate the enrichment scores. Therefore, the calculation of GSVA relies on the background samples.

## Z-score

Gene expressions are standardized over all samples, yielding $z_{ij}$, where $i$ denotes the cases and $j$ refers to the genes belonging to a gene set $J$ [31]. The Z-score for the case $i$ and gene set $J$ is

defined as

$$Z_{iJ} = \frac{\sum_{j=1}^{J} z_{ij}}{\sqrt{|J|}}$$

where $|J|$ refers to the number of genes in get $J$. Obviously, the calculation of Z-score also relies on the background samples.

**2.3.2 Fit pathway scores using LASSO.** The pathway scores converted from gene expression are subsequently fitted using Lasso regression. In the high-dimensional setting with $P > N$, the solution of ordinary least squares is not unique and heavily overfits the data [32]. To handle this issue, regulation is required in this situation. The objective function of Lasso is given by,

$$\hat{\beta}(\lambda) = arg \min_{\beta} \left( \frac{1}{2n} \sum_{i=1}^{n} (y_i - \beta_0 - \sum_{j=1}^{p} X_{ij}\beta_j)^2 + \lambda \sum_{j=1}^{p} |\beta_j| \right) \tag{1}$$

where $\lambda$ is the penalty parameter. The objective function is convex given proper $\lambda$. A property of the Lasso solution is that it does variable selection and shrinkage at the same time because $\beta_i = 0$ after shrinkage is equivalent to variable elimination. The shrinkage parameter $\lambda$ can be chosen empirically with several approaches, such as information criteria (AIC and BIC) and resampling (10-fold cross-validation) [33]. In this study, we optimized the $\lambda$ using 10-fold cross-validation.

## 2.4 Group lasso

The pathway score method is a two-step procedure involving the pathway score calculation and model fitting. In contrast, group lasso is a one-step procedure that can incorporate pathway information. Group lasso is an extension of Lasso that incorporates group structure information. Like lasso, it also has a sparsity nature but achieves the sparsity on the group level, i.e., all variables within a group being zero or non-zero. The objective function is given by,

$$\hat{\beta}(\lambda) = arg \min_{\beta} \left( \frac{1}{2n} \sum_{i=1}^{n} (y_i - \beta_0 - \sum_{p=1}^{P} X_{ij}\beta_p)^2 + \lambda \sum_{g=1}^{G} m_g \|\beta_g\|_2 \right) \tag{2}$$

where $g$ denotes the groups. $m_g = \sqrt{T_g}$, $T_g$ is the cardinality of the $g$-th group. The group lasso reduces to the normal lasso provided $m_g$ is one for all groups.

To mitigate the pathway overlapping issues, we did not include all pathways in the group lasso but selected the 100 most distinct pathways defined by hierarchical clustering. Of course, all pathways will be used for these databases containing less than 100 pathways, such as the hallmark gene set (50 pathways).

## 2.5 Pathway database

Previous studies have accumulated a variety of pathway information based on computation (such as hallmark genes) or prior knowledge (such as WikiPathways) [34]. However, these pathways may be proposed for diverse purposes or diseases, leading to weak association with survival outcomes; Hence, it is critical to select a proper pathway database as prior knowledge. The choice of pathway database can rely on our experience, or we can try multiple pathway databases in internal validation and select the one with the best prediction.

The present studies considered 14 pathway databases derived from MSigDB (http://www.gsea-msigdb.org/gsea/msigdb/collections.jsp), including a Hallmark gene set, a computational gene set (CM), an oncogenic signature gene set (Oncogene), a chemical and genetic

**Table 2. Information about the included pathway databases.**

| Pathways name | Description | Number of pathways |
|---|---|---|
| Hallmark | Hallmark gene sets | 50 |
| Biocarta | BioCarta pathway database | 222 |
| Pid | Pathway Interaction Database | 196 |
| Cgp | Chemical and genetic perturbations | 3023 |
| Wiki | WikiPathways pathway database | 555 |
| Tft.legency | Legend gene set of transcription factor targets | 605 |
| Tft.gtrd | GTRD predicted transcription factor binding sites | 458 |
| Cm | Cancer modules, compiled from a variety of resources such as KEGG, GO, and others. | 427 |
| Gobp | GO Biological Process ontology | 5962 |
| Gocc | GO Cellular Component ontology | 728 |
| Gomf | GO Molecular Function ontology | 1118 |
| Oncogene | Oncogenic signature gene sets | 189 |
| Immune | Immunologic signature gene sets | 4872 |
| Vax | Vaccine response gene sets | 250 |

perturbation pathway (Cgp), two transcription factor targets pathway databases (Tft.legency and Tft.gtrd), two immune signature database (Immune and Vax), three canonical pathways (BioCarta, WikiPathways, and Pid), and three Gene Ontology gene sets (biological process (BP), cellular component (CC), and molecular function (MF)) (Table 2).

## 2.6 Model validation

**2.6.1 Compare predictive models in the internal validation.** We randomly selected 70% of the original data (GSE136324) serving as training data, while the remaining 30% are validation data. Models fitted with the training data were evaluated on the validation data. The process was repeated 100 times. The aim of internal validation was to identify potential models for subsequent external validation. Additionally, the predictive value of various pathway databases can be examined.

**2.6.2 Compare predictive models in external validation.** A model that demonstrated the best performance during internal validation was chosen as the candidate model for subsequent external validation (GSE9782 and GSE2658). The candidate model was then compared against the gene model as well as several previously published models.

## 2.7 Prediction performance measures

We utilized integrated Brier score (IBS) and C-index in both internal and external validation to measure model performance for survival outcomes [35–37]. However, we should be cautious about the interpretation of IBS in external validation because the IBS assumes a consistent baseline hazard between the training and test datasets, yet the baseline hazards may vary across populations (in external validation). For internal validation, we performed a pairwise comparison between models using a corrected resampled t-test for significance [38], whereas for external validation, we exhibited the confidence intervals based on bootstrap resamples.

**IBS score.** Brier score measures the accuracy of predicted probabilities. It gauges the squared distance between the predicted probability and the observed outcomes denoting (0, 1) where 1 means the occurrence of the events, and 0 events do not occur. For the survival data, the Brier score can be used to measure the prediction accuracy for a specific time point, but

insufficient for a range of time. Instead, we can use the integrated Brier score (IBS) that measures the averaged Brier score for an interval of time. To compensate for the information loss due to censoring, we adjusted the Brier score with the inverse of the probability of censoring weighting derived from marginal Kaplan-Meier estimators assuming that censoring is independent of survival and covariates [36]. The censoring weighted Brier score for $t^*$ is

$$BS(t) = \frac{1}{n}\sum\nolimits_{i=1}^{n}\left(I(T_i > t) - \pi(X_i, t)\right)^2 W(t, \hat{G}, X_i) \qquad (3)$$

where $T_i$ is the event time, $I(T_i>t)$ is the indicator of the outcome at time $t$. $\pi(X_i, t)$ is the estimated probability. $W(t, \hat{G}, X_i)$ is the censoring weight, which is given by,

$$W\left(t, \hat{G}, X_i\right) = \frac{I(T_i \le t, T_i \le C_i)}{\hat{G}(T_i)} + \frac{I(T_i > t)}{\hat{G}(t)} \qquad (4)$$

$\hat{G}(t)$ denotes the probability of being uncensored at time $t$.

**C-index.** C-index is a popular metric to measure the concordance between predictive risk scores and observed survival outcomes [35,39]. C-index is simple to understand and more interpretable. It is defined as

$$C = Prob\left(Risk(i) > Risk(j)|T_i < T_j\right) \qquad (5)$$

where $Risk(i)$ is the linear predictor of Cox models. $T_i$ and $T_j$ are the event time. Of note, the pair of cases will not be considered if the order of survival outcomes is unclear because of censoring. Specifically, the following cases will not be considered: i) both cases $i$ and $j$ are censored; ii) $Risk(i)>Risk(j)$ and case $j$ is censored before $T_i$. For the latter, the pair will be used if case $j$ is censored after $T_i$ because it is clear that case $j$ survives longer than case $i$. The C-index should range from 0.5 to 1. A value of 0.5 indicates no discriminating ability, while a value of 1 indicates perfect discriminating ability.

## 2.8 Compare with previous models

We compared our model to four competitor models for MM, including IFM15, UAMS-70, EMC-92, and MILLENNIUM-100. IFM15 was proposed by [40] using the 15 most stable signatures. UAMS-70 was constructed based on GSE2658 using 51 up-regulated and 19 down-regulated signatures [41]. EMC-92 contained 92 signatures proposed based on the gene expression profile of HOVON65/GMMG-HD4 trial (n = 290) [42]. MILLENNIUM-100 was constructed using GSE9782 data, containing the top 100 outcome relevant probes [22]. All models were applied to the MAS5 normalized and log2 scaled data.

Out of the four competitor models, batch effects correction was required for the MILLENNIUM-100, EMC-92, and IFM15 models as they require the absolute expression matrices as input. The new datasets (GSE2658 and GSE9782) were adjusted to be comparable to the training data of these models. The parametric empirical Bayes frameworks, namely ComBat, implemented in the sva package (version 3.46.0) was used for batch effect correction [43]. The UAMS-70 model accepts the relative expression of upregulated and downregulated genes as inputs; hence, batch effects correction is not necessary [44].

## 2.9 Model reporting, implementing, and results reproducing

**Model reporting.** A common challenge in prognostic model studies is the lack of transparent reporting. Many models are only described in articles without being validated in other datasets due to insufficiently detailed model information. Such models become essentially

useless if they are not reported transparently and made easily accessible. The Transparent Reporting of a multivariable prediction model for Individual Prognosis or Diagnosis (TRI-POD) Initiative was developed that comprised a set of recommendations for reporting a prediction model [45]. [46] also proposed guidelines for reporting machine learning models. We improved the reporting of our models following the two guidelines.

**Model implementation.** For future assessment, we formulated all models (not limited to the recommended models) in a package (MMMs) that is accessible on GitHub (https://github.com/ShuoStat/MMMs).

**Missing data imputation.** Models cannot be directly applied to new patients in the presence of missing values. Handling missing values is particularly important for high-dimensional models, as a single missing value can make the predictive model infeasible. Ideally, prediction models should be published with a missing imputation method incorporated.

Our package provided a missing value imputation function based on the K-nearest neighbors, a widely used missing data handling method for high-dimensional data [47]. Specifically, we extracted the K-nearest neighbors of a new patient from the training data and imputed the missing values as the distance-weighted average over K-nearest subjects or genes [48,49]. Here, we searched for K-nearest genes and restricted K to 10, which was the default of the *impute* package in r [50].

To assess the influence of missing values on model prediction, we randomly generated a set of (1%, 2%, 5%, 10%, 20%, and 30%) missing genes in the two external validation data and compared the prediction changes in the presence of missing values.

**Results reproducibility.** The codes for all analyses of this study were provided on GitHub (https://github.com/ShuoStat/MMModels) for result reproducing.

# 3 Results

## 3.1 Internal validation

**Pathway score models were comparable with the gene model.** The performance of the pathway score methods varied across pathway information. In general, the pathway score methods were comparable with the gene model in the internal validation using proper pathway information, such as GSVA in Cgp, ssGSEA in Vax, and Z-score in Gomf (Fig 2, S1 Table). Among the three pathway score methods, ssGSEA models that incorporated Vax and Immune pathways achieved more accurate prediction. It is worth mentioning that the obtained C-index values ranging from 0.55 to 0.60 may not appear particularly convincing in prediction. However, we should be cautious about interpreting the C-index values since it is biased in the small sample size data [51]. This bias does not affect the comparison of different methods as it is consistent across all.

**Vax, Immune, and Hallmark pathways were more predictive for MM.** The predictive power of pathways also differs depending on the methods used for model building. For instance, VAX pathways exhibited favorable prediction using ssGSEA but demonstrated inferior prediction using Z-score or GSVA methods. Regardless of the methods, Vax, Immune, and Hallmark pathways seemed to provide more cancer-relevant information (Fig 2, S1 Table). Hallmark gene sets summarize well-defined biological states or processes. The role of these pathways, such as WNT/β-catenin signaling, TGF-β signaling, and L-6/JAK/STAT3 signaling, have been comprehensively investigated [52–54]. The development of cancer is highly determined by cytotoxic innate and adaptive immune cells. MM, specifically, is a type of cancer that originates in plasma cells, which is an essential component of the immune system [55–57]. Therefore, the immune pathways (including Vax) are highly relevant to the progression of MM.

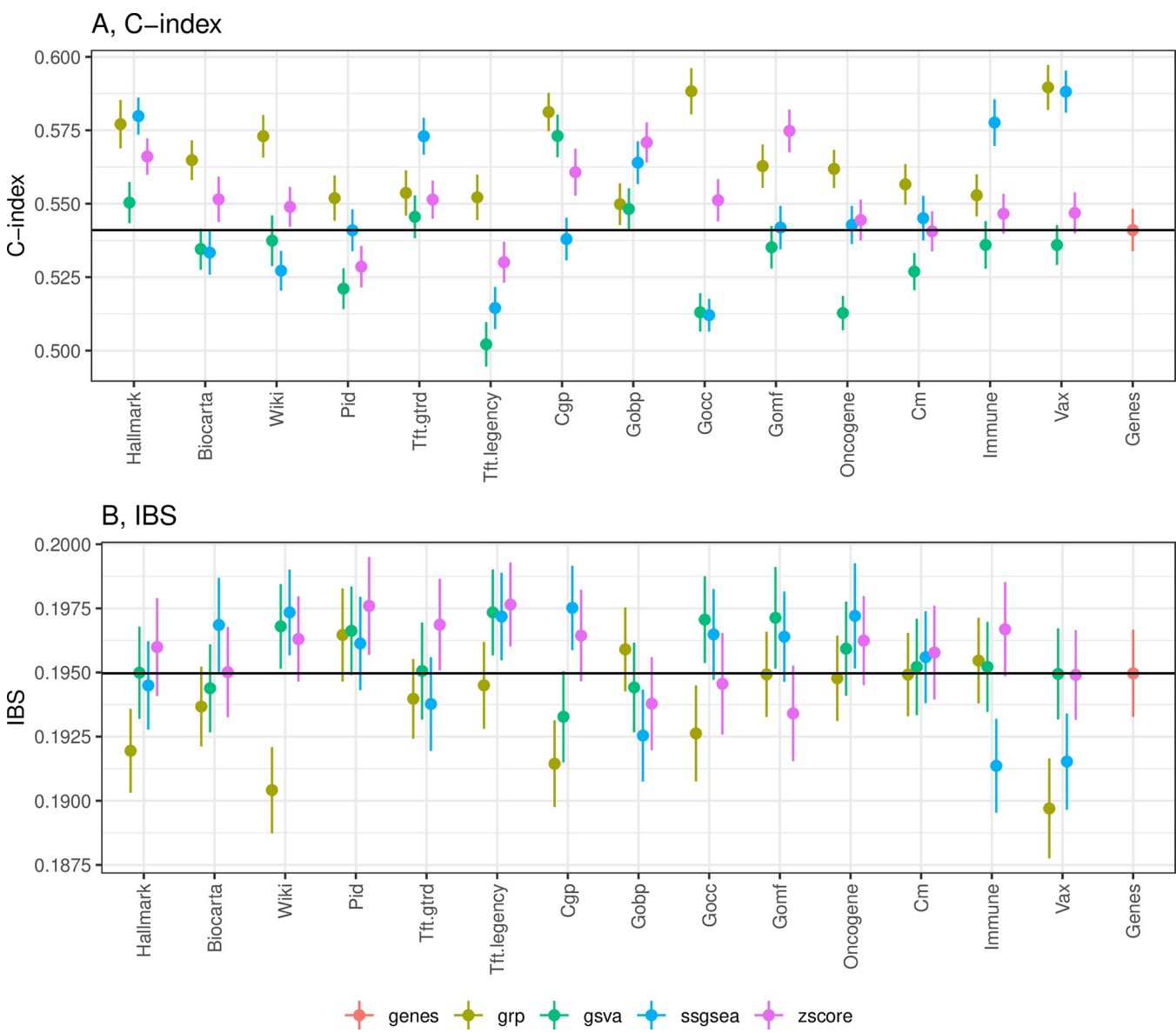

**Fig 2. The prediction performance of pathways-based / gene-based models in internal validation.** Upper, prediction measured with C-index. Lower, IBS. The dots and error bars represent mean ± 1.96*SE.

**Group lasso achieved better prediction.** The advantage of group lasso is apparent in several pathway databases, such as Hallmark, CGP, CGN, GOCC, and VAX. Compared with gene-based models, group lasso selected more variables of small effects, but gained in more accurate prediction and better interpretation of models (Fig 2). The group lasso will select important pathways indicating the survival outcomes, which can serve as a clue for further research on biological mechanisms.

**Select Vax(grp) as a candidate model for external validation.** Based on the internal validation, we selected Vax(grp) for external validation. Vax(grp) demonstrated the highest accuracy in prediction as indicated by C-index and IBS, although no significant difference was observed when compared with some of the competitive models (Figs 2 and S1). Nevertheless, Vax(grp) was still the most promising model based on mean accuracy and therefore was

**Table 3. The prediction performance of selected models in two external data.** Bold, the best one for each measure. Values in the parentheses represent the 95% confidence interval, derived from 1000 bootstrap samples. Vax(grp), Vax(ssgsea), and Genes models were significantly different from each other (Wilcoxon Signed-Rank Test, adjusted P < 0.001). P values were adjusted using Bonferroni method.

| | GSE9782 | | GSE2658 | |
|---|---|---|---|---|
| | **C-index** | **IBS** | **C-index** | **IBS** |
| Vax(grp) | **0.653(0.607,0.700)** | **0.335(0.300,0.369)** | **0.653(0.588,0.722)** | **0.144(0.121,0.160)** |
| Vax(ssgsea) | 0.632(0.581,0.681) | 0.347(0.322,0.394) | 0.604(0.536,0.663) | 0.151(0.133,0.174) |
| Genes | 0.618(0.569,0.668) | 0.349(0.314,0.385) | 0.625(0.554,0.694) | **0.144(0.124,0.163)** |

chosen for further comparison in external data. Additionally, we included a pathway score model, namely Vax(ssGSEA), to further enhance our understanding of this approach on the test data. All models were then retrained using the entire training dataset and compared with the gene-based model.

## 3.2 External validation

**Vax(grp) outperformed the gene model.** Table 3 presents the prediction performance of the selected candidate models in two external data (GSE2658 and GSE9782). Vax(grp) outperformed the other two models in both data, based on C-index, but was comparable to the gene model measured by IBS in GSE2658. Vax(ssgsea) is superior to gene model in GSE9782, but inferior in GSE2658. Statistical tests indicated that all three models were significantly different from each other, with Vax(grp) demonstrating superior performance (Table 3). Collectively, Vax(grp) remained to be the most competitive model in external validation.

## 3.3 Vax(grp) outperformed the previous models

Calculating IBS was not feasible for the previous models due to the absence of their baseline hazards. Hence, we only presented the C-index of the three data in Table 4. Vax(grp) model in GSE136324, UAMS-70 in GSE2658, and MILLENNIUM-100 in GSE9782 demonstrated the best performance in the corresponding data. However, they should not be considered for comparison because these models were trained with corresponding data (Table 4, S2 Fig). Vax(grp) consistently outperformed all four existing well-known models in all three data (GSE136324 is the training data) (all P < 0.001) (Table 4). The proposed models also exhibited a favorable capacity in differentiating the high- and low-risk groups if patients were categorized based on the medians of linear predictors (S2 Fig). EMC-92 was another competitive model that performed well in GSE2658 and GSE9782, but not GSE136324.

In general, while all models showed strong performance in their training data, their performance significantly deteriorated when applied to other datasets, occasionally resulting in a C-index of less than 0.6. However, Vax(grp) appeared to be more reliable even when used with

**Table 4. Compare the proposed model with previous models using C-index.** *, data was the training data of that model. Bold, the best one for each data. Values in the parentheses represent the 95% confidence interval, derived from 1000 bootstrap replicates. Vax(grp) was significantly different from all competitor models across three data (Wilcoxon Signed-Rank Test, all P < 0.001).

| | GSE136324 | GSE2658 | GSE9782 |
|---|---|---|---|
| EMC-92 | 0.567(0.520, 0.610) | 0.634(0.562, 0.706) | 0.629(0.578, 0.680) |
| UAMS-70 | 0.560(0.514, 0.602) | 0.658(0.588, 0.726)* | 0.606(0.553, 0.656) |
| IFM15 | 0.508(0.465, 0.553) | 0.568(0.507, 0.628) | 0.563(0.510, 0.615) |
| MILLENNIUM-100 | 0.582(0.538, 0.627) | 0.575(0.506, 0.640) | 0.682(0.633, 0.726)* |
| Vax(grp) | 0.854(0.829, 0.878)* | 0.653(0.588, 0.722) | 0.653(0.607, 0.700) |

different datasets. The possible reason is that Vax(grp) selects more variables (736 variables in 20 pathways) (S2 Table), leading to a reduced variance of prediction.

### 3.4 Model reporting

For transparent reporting, we summarized the important elements of the models in Tables 5 and S3 following TRIPOD model reporting structure.

### 3.5 The effects of missing values on model prediction accuracy

We integrated a missing value imputation method in the model package to enhance its usability. With the imputation method, a certain proportion of missing values would not deteriorate the prediction performance. For both GSE2658 and GSE9782, the C-index for all three methods remained almost unchanged with less than 2% missing values, and experienced a slight increase in variance if the percentage of missing values was 5% (Fig 3). The ssGSEA model was more resistant to missing values because ssGSEA is a rank-based method. The gene model exhibited a larger variance than Vax(grp), especially for a large percentage of missing. Group lasso selected more variables with small effects, thereby reducing the influence of individual variables. In contrast, the gene model selected fewer variables, some of which had larger effects. The missing of these important variables in the gene model can severely deteriorate the prediction. In the model package, we default that 5% of missing values would be allowed if imputation is used.

### 3.6 computational complexity

The challenge of implementing these models in clinical practice lies in the data preparation procedure. For models based on pathway scores, data preparation involves batch effects adjustment, missing data imputation (if necessary), and pathway score computation. For models based on group lasso, data preparation includes batch effects adjustment and missing data imputation (if necessary).

In general, without missing values, prediction (including batch effect correction) can be achieved within seconds (Table 6). However, the presence of missing values can extend the process to several minutes. Overall, compared to the time required to generate gene expression data for new patients, which may take days to weeks, the time required for running our bioinformatic pipeline is negligible and therefore does not introduce significant hurdles for clinical translation.

## 4 Discussion

Predictive models are important to stratify patients for therapeutic decision-making in case patients with different survival risks are managed in a similar manner [58]. The present study investigated two model-building strategies that can incorporate pathway information. Their prediction performance varied by pathways, but immune information, including the Vax, seemed to be more predictive for MM. Between the two strategies, group lasso was better off for the majority of pathways, and achieved the best prediction when combined with VAX pathways. This model was still competitive even when applied to the population with different backgrounds and treatments (external validations).

The challenge for pathway score methods is that they cannot capture sufficient information, which may account for the inferior performance compared with group lasso. Among the three pathway score methods, ssGSEA seems to be more reliable probably because its calculation

**Table 5. A brief report on the proposed predictive model (Vax(grp)) for multiple myeloma.**

| **1, Title** | |
|---|---|
| A prognostic model for multiple myeloma (MM) incorporating pathway information | |

| **2, Abstract** | |
|---|---|

**Objectives**
Construct predictive models for MM by incorporating pathway information.
**Methods**
We considered two strategies for model building that could take pathway information into consideration: pathway score method and group lasso. These two strategies were applied to microarray data for MM (GSE136324) and the yielded models were compared with a gene-based model in internal validation. A candidate model fitted with group lasso using Vax pathway (Vax (grp)) information exhibited the best prediction performance. This model was further validated with two external data (GSE9782 and GSE2658).
**Conclusion**: The Vax (grp) model outperformed the gene-based model and previous existing models in prediction accuracy. The model was implemented in an R package, accessible on GitHub (https://github.com/ShuoStat/MMMs) for future evaluation and comparison.

| **3, Key elements of model building** | |
|---|---|
| Source of data | GSE136324, See Table 1 for more detailed information |
| Participants | Patients newly diagnosed as MM were enrolled. These patients were treated with Total Therapy 3. In cases where patients had repeated measurements, we utilized the first measurement for analysis. |
| Outcome | Overall survival: follow-up time ranged from 0–174 months |
| Predictors | Data was obtained from Affymetrix Human Genome U133 Plus 2.0 Array platform. Data were normalized with MAS5.0 and log2 transformed. See Section 2.1 for more data preprocessing procedure. Eventually, a total of 13,039 genes were retained for model building. |
| Sample size | 436 cases with 191 (43.8%) observed events |
| Model building | i) We selected the 100 most distinct pathways from the Vax pathway database based on hierarchical clustering.<br>ii) Subsequently, we built models using group lasso, where the group structure was defined by these 100 pathways. |
| Competing models | 1, Models fitted with the same training data (GSE136324)<br>i) Models using pathway score strategy.<br>ii) Gene model<br>2, Previous models<br>i) IFM15<br>ii) UAMS-70<br>iii) EMC-92<br>iv) MILLENNIUM-100 |
| Validation | Both internal and external validation were used.<br>1, Internal validation: original data (GSE136324) was divided into 70% training data and 30% validation data. The process was repeated 100 times.<br>2, External validation: The developed model was further validated using two external datasets, namely GSE9782 and GSE2658. |
| Performance measures | 1, C-index; 2, Integrated Brier Scores (IBS). |
| Missing data | No missing value in model building procedure. |
| Model specification | 1, To implement the proposed model, the new data should be obtained from the Affymetrix Human Genome U133 Plus 2.0 Array platform, preprocessed with MAS5.0 normalization and log2 transformation.<br>2, The new data should be adjusted for batch effects using the training data (GSE136324) as the reference.<br>3, A missing value imputation method based on K-nearest neighbors was provided along with the model. However, it is important to ensure that datasets with missing values do not exceed 5% of the total data. |
| Implications | Models were implemented in an R package, accessible on GitHub (https://github.com/ShuoStat/MMMs) |
| Limitations | 1, clinical information was not taken into consideration. 2, the proposed model selected a greater number of variables, resulting an increased cost for practical application. 3, we did not provide cutoff to categorize the high- and low-risk groups. 4, limited to Lasso regression. However, exploration of more advanced machine learning methods is required. |

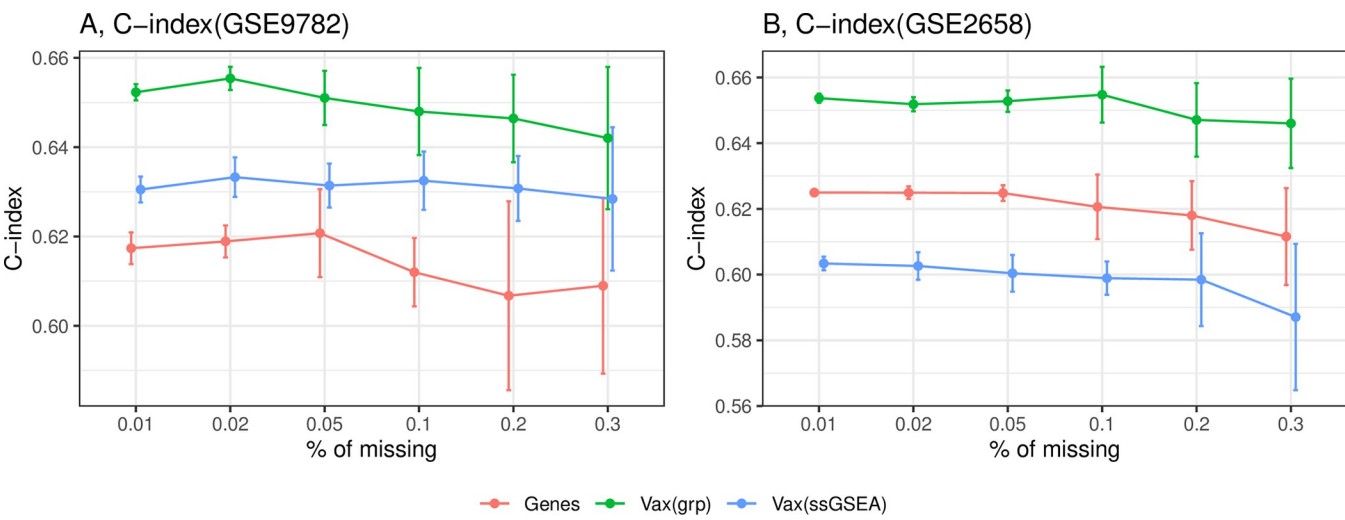

**Fig 3. The influence of missing values on prediction.** X-axis is the percentage of random missing values.

merely relies on gene ranks without the adjustment of background samples. In contrast, the calculation of GSVA and Z-score relies on background samples.

Pathway-based modeling strategy predicts prognostics using the activities of biological functions. This approach underscores the idea that cancer development involves the disorder of relevant biological functions rather than the alteration of a few specific genes. The resulting model not only serves as a prognostic tool but also highlights the functions relevant to the disease, owing to the sparsity property of the Lasso regression. For mechanism research, instead of using a single model to identify reliable pathway information, we recommend conducting stability analysis on these proposed models and prioritizing these frequently selected pathways for subsequent research [59,60].

Regarding the selection of prior knowledge, we prefer the knowledge-based database over the computation-originated database. For example, Gene Ontology (GO) is founded on previous research with more solid evidence. In contrast, the computational gene set can be considered weak evidence.

Compared with the gene-based model, the proposed Vax(grp) was more accurate in prediction and interpretable. The gene-based models select individual genes as predictors. Due to the collinearity of the gene expression profile (many genes show high correlation with each other) and the regularization (e.g. Lasso penalty) during feature selection, the genes selected in the model are not readily interpretable as genes correlated with the selected ones may be randomly

**Table 6. Model computation cost when applied to new data (unit, seconds).**

|  | GSE9782 | | | GSE2658 | | |
|---|---|---|---|---|---|---|
|  | **Vax(grp)** | **Vax(ssgsea)** | **Genes** | **Vax(grp)** | **Vax(ssgsea)** | **Genes** |
| Batch Effects Correction | 0.884 | 0.885 | 0.867 | 1.196 | 1.656 | 1.667 |
| Path Score Calculation | / | 2.629 | / | / | 4.363 | / |
| Prediction | 0.002 | <0.001 | <0.001 | 0.004 | <0.001 | <0.001 |
| Missing Value Imputation | 254.421 | 265.482 | 253.443 | 506.587 | 497.131 | 499.474 |

Notes: Models were executed using R Software (version 4.3.0) on the Windows 11 (x64) operating system. The computer was equipped with an AMD Ryzen 9 7950X 16-Core Processor and four 16GB RAM modules. No parallel running was used. Additionally, 1% of missing values were randomly generated.

dropped by the model, but they may have important biological functions. On the other hand, the pathway-based models first summarize the individual gene expression changes on pathway activity level, which can be considered as a pathway-guided dimension reduction and denoise process. Therefore, the pathway-based models are more stable and interpretable than the gene-based models. For example, in our study, the two most predictive models are those based on immune-related pathways, which is consistent with the fact that Multiple Myeloma is a cancer of plasma cells, which produce antibodies and therefore are a critical component of the adaptive immune system [61,62]. The proposed model Vax(grp) selected 20 different pathways from the Vax pathway database (S2 Table). Vax database was derived from the Human Immunology Project Consortium, describing human transcriptomic immune responses to vaccinations. Previous studies have documented the association between virus response and MM [63,64]. Functions involved in the hepatitis B virus response were predictors in our model. Many studies have demonstrated that virus infection increases the risk of multiple myeloma [65–68]. The other selected pathways were also investigated by previous studies, including vaccine responses to HIV, Measles, Shingles, Human papillomaviruses (HPV), Tuberculosis, and Influenza [69–78]. These selected pathways provide clues guiding future research.

Although numerous models have been proposed, few have transitioned into clinical applications. This gap largely arises from insufficiently transparent reports and a lack of neutral comparison. For this sake, we improve the reproducible documentation of our models, including information on the patient populations, training data, validation data, and model coefficients. However, we acknowledge that significant work remains to be done before these models can be implemented in clinical settings or impact therapeutic decision-making. First of all, more independent test datasets are needed to validate the robustness and generalizability of our current model. Secondly, our model can also benefit from further optimization, such as including clinical parameters as additional predictors as well as implementing other machine learning/AI methods for deriving pathway scores and prediction. Most importantly, any models still need to undergo rigorous clinical trials before the approved by regulatory bodies, such as the FDA in the United States or the EMA in Europe, and the eventual applications in routine clinical practice [79,80].

The primary contribution of this study is our proposal to incorporate existing knowledge into model building. Data-driven model building, such as gene models, is challenging due to the need for a large sample size. However, it is difficult to obtain enough samples, especially in high-dimensional setups. Building pathway models is guided by existing knowledge, which can reduce the sample size requirement and allow us to build on previous research findings, rather than starting from scratch.

We are also aware of the limitations of the current study. Firstly, we only covered a limited number of pathway-scoring methods and prediction models. For example, we only used Lasso regression for predicting clinical outcomes due to its simplicity and interpretability. However, other machine learning models, such as Random Forest, XGBoost or deep learning models may further improve the prediction accuracy. Better pathway scoring methods, such as those considering the direction of differential expression, may also benefit our model. Secondly, while our model proposed Vax, a computational pathway database of vaccination related gene expression changes, as the best source of pathway knowledge, the evidence of biological relevance is currently unclear, and awaits experimental validation. Last but not least, we only tested our approaches in Multiple Myeloma. To make our method more generalizable, a border benchmark study in the future is required to test whether the pathway-based models improve patient stratification in our cancer types and which knowledge database should be used for a specific disease. Including disease-specific clinical parameters, such as tumor stages, genomic background, and so on, may provide additional power to our prediction model.

Nevertheless, our current study provides a valuable guidebook for such benchmark studies in the future.

## Supporting information

**S1 Table. The prediction performance of different methods in internal validation.** The values are the mean and standard deviation of 100 duplicated runs.
(DOCX)

**S2 Table. The selected pathways of the Vax(grp) model.**
(DOCX)

**S3 Table. TRIPOD Checklist: Prediction Model Development.**
(DOCX)

**S1 Fig. Statistical significance for pairwise comparisons.** P-values were obtained using a corrected resampled t-test and were not adjusted for multiple testing. The lower triangular part of the matrix represents the significance for IBS, while the upper triangular part represents the significance for the C-index.
(DOCX)

**S2 Fig. Kaplan-Meier Plot. The survival curve of the high and low risk groups predicted by different models.** The median of linear predictors was used as the cutoff. *, the model was trained with that data.
(DOCX)

## Acknowledgments

We appreciate the contributors of the data used in this research, including but not limited to Samuel Danziger (GSE136324), Shaughnessy Jr. John (GSE2658), Bryant B, and Mulligan G (GSE9782).

## Author Contributions

**Conceptualization:** Shuo Wang, ShanJin Wang, Wei Pan, Junyan Lu.

**Data curation:** Shuo Wang, Wei Pan, YuYang Yi.

**Formal analysis:** Shuo Wang, Wei Pan.

**Funding acquisition:** ShanJin Wang, Wei Pan, Junyan Lu.

**Investigation:** Shuo Wang, ShanJin Wang, Wei Pan, YuYang Yi, Junyan Lu.

**Methodology:** Shuo Wang, ShanJin Wang, Wei Pan, Junyan Lu.

**Project administration:** YuYang Yi.

**Resources:** ShanJin Wang, Junyan Lu.

**Software:** Shuo Wang, Junyan Lu.

**Supervision:** ShanJin Wang, Junyan Lu.

**Validation:** ShanJin Wang, Wei Pan, YuYang Yi, Junyan Lu.

**Visualization:** Shuo Wang, Junyan Lu.

**Writing – original draft:** Shuo Wang, ShanJin Wang, Wei Pan, YuYang Yi.

**Writing – review & editing:** Shuo Wang, ShanJin Wang, Wei Pan, Junyan Lu.

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
