## [Decision Letter · Decision Letter 0]

8 Jun 2024

Dear PhD Lu,

Thank you very much for submitting your manuscript "Construct Prognostic Models of Multiple Myeloma with Pathway Information Incorporated" for consideration at PLOS Computational Biology.

As with all papers reviewed by the journal, your manuscript was reviewed by members of the editorial board and by several independent reviewers. In light of the reviews (below this email), we would like to invite the resubmission of a significantly-revised version that takes into account the reviewers' comments.

The pathway-based approach was found interesting by all the reviewers, while pointing out shortcomings that, if addressed, may significantly strengthen the paper. Please address these suggestions as best as you can.

Reviewer 2 noted that a discussion related to the computational complexity of the presented model. and how it compares to existing models, considering its accuracy, is missing. This is an important point to address, as it has a direct impact on clinical translation, as noted also by Reviewer 3.

We cannot make any decision about publication until we have seen the revised manuscript and your response to the reviewers' comments. Your revised manuscript is also likely to be sent to reviewers for further evaluation.

Sincerely,

Gianfilippo Coppola

Guest Editor

PLOS Computational Biology

Mark Alber

Section Editor

PLOS Computational Biology

Reviewer's Responses to Questions

**Comments to the Authors:**

Reviewer #1: The manuscript provides insights into the use of pathway information for developing prognostic models in Multiple Myeloma. However, I have three minor points for revision:

1) Clarity on Batch Correction Application: Please specify where batch correction was applied in your analyses. Your manuscript mentions adjusting batch effects for the training data of competitor models, when applicable (Page 15, Line 15). Clarifying exactly where batch correction was applied (and where it was not) would greatly aid in understanding the methodological rigor and ensuring the robustness of your comparative analysis.

2) Pathway Scoring Justification: Could you provide a more detailed motivation for your choice of pathway scoring methods? While the comparison with gene-based models is clear, understanding the specific rationale behind selecting these pathway scoring methods would enhance the context and depth of your study.

3) The sentence in Page 5, Line 9 seems to have grammatical issues

Best regards

Reviewer #2: In this paper, the authors explore novel prognostic modeling approaches for Multiple Myeloma (MM), focusing on the utilization of pathway information rather than conventional gene-level data. Two primary strategies are assessed: the pathway score method and group lasso with pathway information. The study tests various methods for calculating pathway scores and integrates this information into predictive models, using microarray data (GSE136324) for MM. The best-performing model, based on group lasso incorporating Vax pathway information (Vax(grp)), demonstrates superior predictive accuracy compared to gene-based models and previously published approaches, both in internal and external validations. Despite the manuscript presenting high levels of accuracy, a good level of English and providing a valuable contribution to current research, it needs some improvements:

- Improve the introductory section of the manuscript by highlighting and listing both the “contributions'” points of this research and the work's limitations;

- While the study presents promising results, the generalizability of the findings to other types of hematological diseases remains to be explored. Please discuss them in the possible limitations of the work or as possible future tasks;

- Considerating the complexity of Pathway Analysis, please discuss how the complexity of these models may affect their translation from research to clinical practice, considering the computational load that may be placed on clinical systems by offering potential solutions or areas for future development that may help reduce the computational burden, such as algorithmic improvements or the use of more efficient computing frameworks. So, explain better the complexity of the proposed framework, which might make it difficult to implement in real-world scenarios;

- Please to include specific statistical tests to quantify the effectiveness of the proposed method and simulation;

- This study could benefit from comparisons with the latest deep learning or machine learning techniques in prognostic modeling. Please to discuss them.

- Given the complexity of pathway-based models, the study could delve deeper into the interpretability of these models, which is crucial for clinical acceptance and application. How do the authors overcome and discuss that difficulty?

- The transition from a research setting to clinical implementation requires further validation, especially in terms of how these models might influence therapeutic decision-making in real-world scenarios. Please discuss that;

- The "Conclusions" section should be expanded to offer a more comprehensive view and analytical perspective on future prospects by considering and adding a comparative analysis.

Reviewer #3: The paper from Wang et al is of interest and describe a new method to infer and interpret the pathway analysis and how they can explain cancer development and biology.

The paper is well written and despite the high complexity of the applied model it is easy to understand.

I have only a concern based on the idea that this model could be applied for decision making and also could outperformed compared with the previous models.

This could be right from a technical point f view, which is the scope of the work, but for really say that outperformed it would be nice if the authors could provide some survival predictions based on they new model.

The knowledge of how a cancer work (deregulated pathways) is absolutely more important rather than the knowledge of a some gene deregulation. Nevertheless, the application of other scores, even if gene-based, has clinical and prognostic impact. Is also the case for the author's model?

**Have the authors made all data and (if applicable) computational code underlying the findings in their manuscript fully available?**

Reviewer #1: Yes

Reviewer #2: Yes

Reviewer #3: Yes

PLOS authors have the option to publish the peer review history of their article (what does this mean?). If published, this will include your full peer review and any attached files.

Reviewer #1: No

Reviewer #2: **Yes: **Giovanni Cicceri

Reviewer #3: No
---

## [Decision Letter · Decision Letter 1]

28 Aug 2024

Dear PhD Lu,

We are pleased to inform you that your manuscript 'Construct Prognostic Models of Multiple Myeloma with Pathway Information Incorporated' has been provisionally accepted for publication in PLOS Computational Biology.

Best regards,

Gianfilippo Coppola

Guest Editor

PLOS Computational Biology

Marc Birtwistle

Section Editor

PLOS Computational Biology

Reviewer's Responses to Questions

**Comments to the Authors:**

Reviewer #1: Dear Authors,

Thank you for your thorough responses and the revisions made to the manuscript. After reviewing the updated version, I am pleased to see that my previous comments and concerns have been addressed satisfactorily.

I have no further comments at this time and believe the manuscript has been improved.

Best regards

Reviewer #2: The manuscript has been revised to enhance its clarity and organization of findings. All suggested sections have been refined to elevate the paper's quality. Additionally, the discussion has been updated and the conclusions strengthened.

Reviewer #3: I thank the authors to have substantially accomplished to my previous comments.

Nevertheless a multivariate analysis to demonstrate a clinical significance and a real ability to overcoming other risk stratification methods would still be useful.

**Have the authors made all data and (if applicable) computational code underlying the findings in their manuscript fully available?**

Reviewer #1: Yes

Reviewer #2: None

Reviewer #3: Yes

PLOS authors have the option to publish the peer review history of their article (what does this mean?). If published, this will include your full peer review and any attached files.

Reviewer #1: No

Reviewer #2: **Yes: **Giovanni Cicceri

Reviewer #3: **Yes: **Matteo Claudio Da Viá

---

## [Editor Report · Acceptance letter]

4 Sep 2024

PCOMPBIOL-D-23-01610R1 

Construct Prognostic Models of Multiple Myeloma with Pathway Information Incorporated

Dear Dr Lu,

I am pleased to inform you that your manuscript has been formally accepted for publication in PLOS Computational Biology. Your manuscript is now with our production department and you will be notified of the publication date in due course.

With kind regards,

Zsofia Freund
